# Safety and Immunogenicity of a New Rotavirus-Inactivated Vaccine in the Chinese Adolescent Population: A Randomized, Double-Blind, Placebo-Controlled Phase I Clinical Trial

**DOI:** 10.3390/vaccines13040369

**Published:** 2025-03-30

**Authors:** Yan Liu, Guangwei Feng, Jinyuan Wu, Xinling Liu, Jing Pu, Yanxia Wang, Wangyang You, Na Yin, Shan Yi, Jiebing Tan, Xiaochen Lin, Lili Huang, Jiamei Gao, Qingchuan Yu, Qiumeng Tong, Yong Zhang, Rong Chen, Xiaoqing Hu, Jun Ye, Xiangjing Kuang, Yan Zhou, Zhongyu Hu, Dongyang Zhao, Hongjun Li

**Affiliations:** 1National Institutes for Food and Drug Control, Beijing 102629, China; liuyan418@nifdc.org.cn (Y.L.); gaojiamei@nifdc.org.cn (J.G.); yqc@nifdc.org.cn (Q.Y.); tongqiumeng@nifdc.org.cn (Q.T.); zhangyong@nifdc.org.cn (Y.Z.); 2State Key Laboratory of Drug Regulatory Science, Beijing 102629, China; 3Henan Center for Disease Control and Prevention, Zhenzhou 450016, China; vacfeng@163.com (G.F.); wangyanxia99@163.com (Y.W.); dsrt12345@163.com (W.Y.);; 4Institute of Medical Biology, Chinese Academy of Medical Science & Peking Union Medical College, Yunnan Key Laboratory of Vaccine Research and Development on Severe Infectious Disease, Kunming 650118, China; wujinyuan@imbcams.com.cn (J.W.); liuxl_imbcams@163.com (X.L.); pujing@imbcams.com.cn (J.P.); yanyan_850@163.com (Y.Z.)

**Keywords:** rotavirus-inactivated vaccine, safety, immunogenicity

## Abstract

Background: We performed a phase I experiment in a healthy teenage population in Sui County, Henan Province, China. The trial was randomized, double-blind, and placebo-controlled. Methods: Ninety-six adolescents were randomly assigned in three groups (high-dose, medium-dose, and low-dose) to receive a dose of the vaccine or the placebo. The patients were monitored for adverse events (AEs) for up to 30 days after each dose of the vaccine and for up to 6 months after all doses of serious AEs (SAEs). All observed AEs and SAEs were reported. Microneutralization assays were used to measure geometric mean titers (GMTs) and seroconversion rates for neutralizing antibodies. IgA and IgG antibodies specific to the rotavirus were detected. Results: The rates of total AEs in these groups were 8.33%, 37.50%, 12.50%, and 4.17%, respectively. The neutralizing antibody test revealed that the teenage groups with low, medium, and high doses of the vaccine had geometric mean titers of 424.32, 504.63, and 925.45, respectively, at 28 days following complete vaccination. The GMT of serum IgG at final immunization was 6501.86, 6501.82, and 10,173.3, in the low-dose, medium-dose, and high-dose groups, respectively. The GMT of serum IgA at final immunization was 2733.64, 2233.29, and 3596.66 in the low-dose, medium-dose group, and high-dose groups, respectively. Conclusions: The majority of adverse events (AEs) were deemed Grade 1 or 2, suggesting that the vaccine’s safety profile is suitable for healthy adolescents. For the primary immunogenicity endpoints, a preliminary examination of the GMTs and the positive transfer rate of neutralizing antibodies in the different experimental groups revealed that, in adolescents aged 6–17, the high-dose group displayed significantly higher levels of neutralizing antibodies compared to the medium- and low-dose groups. Adolescents had few side effects from the new inactivated rotavirus vaccination, and it elicited an immune response.

## 1. Introduction

Rotavirus gastroenteritis is common global phenomenon [1,2,3,4], affecting individuals of all age groups, but with a particularly high prevalence in children under five years of age [5]. Almost every child will be infected with rotavirus gastroenteritis at least once between the ages of 3 and 5 [6,7]. The infection rate is higher in children living in diaspora groups, although there is no statistically significant difference between the sexes [8]. In addition to causing diarrhea, extraintestinal symptoms caused by rotavirus infection are also of concern [9,10]. The risk of rotavirus infection to children’s growth and health cannot be ignored. 

There are currently seven rotavirus vaccines licensed and marketed worldwide, and four have successively received WHO prequalified status [11,12,13,14,15,16,17,18]. The safety profile of rotavirus vaccines approved for use is favorable, with the main symptoms being pyrexia, vomiting, diarrhea, irritability, and other general reaction symptoms [11,19,20,21]. These symptoms are typically transient and rarely result in serious adverse events. At present, three rotavirus vaccines are available on the Chinese market (LLR, LLR3, and RV5) [15]; however, rotavirus vaccination coverage is relatively low in Asia.

In China, rotavirus gastroenteritis historically occurs year-round in kids under five, with a clear seasonal distribution, where the peak incidence of the disease usually occurs from November to February [22,23,24,25]. After the rotavirus vaccine became available, the start and end months of the peak rotavirus gastroenteritis epidemics in cities in the southern provinces of China showed a tendency to shift; the detection rate of rotavirus during the peak season decreased, and compared to the northern provinces, the southern provinces had a far lower rotavirus detection level [22,24]. Despite the improvement that has been observed with the availability of rotavirus vaccines, there is considerable room for enhancement in terms of vaccine types, vaccine components, and protective effects. For the purpose of preventing rotavirus infections in children, the creation of a deactivated rotavirus vaccine would pave the way for the introduction of a new, secure, and easily administered vaccination alternative.

The rotavirus outer capsid proteins VP7 and VP4 have different antigenicities; the glycosylated VP7 protein is known as the G serotype or genotype, while the protease-sensitive VP4 protein is known as the P serotype or genotype. With the development of genetic identification techniques, genotyping became a widely used technique for rotavirus identification [26]. Currently, the Rotavirus Classification Working Group classifies rotaviruses into 42 G serotypes/genotypes (G1–G42) and 58 P serotypes/genotypes (P[1]–P[58]). Rotaviruses that infect humans have 18 G genotypes and 19 P genotypes. The major rotavirus genotypes that infect humans worldwide include G1P[8], G2P[4], G3P[8], G4P[8], and G9P[8] [27]. The Institute of Medical Biology, Chinese Academy of Medical Science, developed a novel inactive rotavirus vaccine that was used in this study. Vaccination was based on the G1P human wild-type rotavirus [8]. It was grown in Vero cells, and then formaldehyde was used to purify and inactivate it. An aluminum hydroxide adjuvant was then added for production. The target population for inactivated rotavirus vaccine is infants and children under the age of 5 years. Phase I clinical trials of inactivated rotavirus vaccine have included adults, adolescents, infants and children. In a previous study, we systematically evaluated the safety and immunogenicity of different doses of the vaccine in a population of healthy adults [28]. In this study, safety and immunogenicity were evaluated in the adolescent population to evaluate the safety of different doses of the vaccine and to provide a database for the subsequent design of the immunization program.

## 2. Materials and Methods

### 2.1. Vaccines

The Chinese Academy of Medical Sciences’ Institute of Medical Biology developed and manufactured the inactivated rotavirus vaccines. The G1P[8] genotype of human rotavirus strain ZTR-68 was cultivated, viral fluid was recovered, the virus was purified, inactivated, and finally created by adding an aluminum hydroxide adjuvant. The Vero cells were infected with this strain of rotavirus. A somewhat milky white suspension was what the vaccine looked like. Syringes were pre-filled with a single dose of the vaccine (0.5 mL). The vaccine was produced with three different dosage specifications of the rotavirus-inactivated antigen: 80EU, 160EU, and 320EU, respectively.

### 2.2. Clinical Trial Design

A double-blind, randomized, placebo-controlled experiment was carried out as part of the phase I study. Low, medium, and high doses were used to divide the vaccination groups. In each dosage group, there was a 3:1 ratio of vaccination to placebo. A vaccine-to-placebo ratio of 1:1 was achieved by combining the placebo populations from all three dosage groups, which allowed for easier examination of the vaccine’s effectiveness. There were 32 instances in each dosage group for the vaccination and placebo groups. In order to guarantee the safety and uniformity of the clinical study, a Data Safety Monitoring Board (DSMB) was formed to evaluate the risks.

Ensuring the blinding process’s integrity required that only personnel not involved in the blinding procedure could access the correspondence between the subject number and the vaccine number. Once the assignment was finished, the correspondence was sealed. Nobody in the clinical experiment knew about the mapping link between the subjects’ numbers and the vaccines’ numbers.

To determine whether the program’s requirements had been fulfilled, on the seventh day after the first immunization, the findings of the initial security review were evaluated. This included checking to see whether any participants had Grade 3 severe adverse responses, or if the percentage of vaccinated individuals reporting such reactions did not surpass 15%. It also included looking for cases of major unexpected suspected adverse reactions or vaccine-related Grade 4 severe adverse reactions. At least 30 min after each dose of vaccination, the subjects were monitored on-site. Prior to each dosage of the vaccine and on the fourth day after immunization, the participants (ranging in age from 6 to 17) in these groups were asked to undergo blood biochemistry, as well as regular blood and urine testing. All participants continued to be observed for adverse events from 8 to 30 days and records were kept. The participants were required to continue follow-up up to 6 months after full vaccination, with only serious adverse events (SAEs) recorded. The International Council for Harmonisation of Technical Requirements for Pharmaceuticals for Human Use (ICH) has published numerous guidelines, including the ICH E6 Good Clinical Practice (GCP) guideline, which provides standards for reporting of clinical trials and definition of AE and SAEs [29].

GMT values and positive transfer rates twenty-eight days following 2 injections of the vaccine were the main endpoints of humoral immunogenicity. The vaccine was designed to neutralize rotavirus. The GMT values and serum positive rates for anti-rotavirus immunoglobulin G (IgG) and IgA antibodies were the exploratory objectives related to humoral immunogenicity, measured 28 days after the second dosage of the vaccine.

Twenty-eight days after the second dosage, serum was collected, and the presence of neutralizing antibodies, and the levels of IgA and IgG antibodies, were determined. Additionally, the geometric mean titer (GMT) values were calculated. The positivity of neutralizing antibodies was defined as a negative result for the antibody in question before the administration of immunization, followed by a positive result after immunization (where an antibody potency lower than 1:8 was considered negative and a value of 1:8 or above was considered positive). Alternately, a positive antibody test before immunization and a subsequent fourfold rise in antibody levels relative to baseline values were considered to constitute positivity. There were three ways in which IgG antibody positivity was defined: first, if the pre-immunization antibody was negative but the post-immunization antibody was positive (with an antibody potency of ≤1:16 being considered negative and a value >1:16 being considered positive); second, if the pre-immunization antibody was positive but the post-immunization antibody was four times higher than the baseline value; or third, if the pre-immunization antibody had a fourfold increase in the post-immunization antibody levels compared to the baseline value. When the pre-immunization antibody was negative and the post-immunization antibody was positive, it was considered that the serum IgA antibodies had been transferred successfully (with a rotavirus IgA level of <1:8 indicating a negative result, and a value ≥1:8 indicating a positive result), or the pre-immunization antibody showed a positive result, and the antibody level increased fourfold following vaccination compared to the baseline value. (ClinicalTrials.gov number, NCT04626856)

### 2.3. Determination of Neutralizing Antibody Titers

The Institute of Medical Biology of the Chinese Academy of Medical Sciences worked on preparing the indicator virus, which was then kept at a temperature of −60 °C. To activate the virus, 20 µg/mL of trypsin was added to the viral solution, and then the combination was put in a tub of water at 37 °C for 60 min. After that, we used a 96-well plate to examine and neutralize the virus at 37 °C for 2 h by mixing 50 µL of viral indicator solution (100 CCID_50_/well, neutralization unit) with an equivalent quantity of serially diluted serum. Two wells were filled for every serum dilution, and the plates were incubated at 37 °C with 5% CO_2_ for seven days. After incubation, the plates were placed in a low-temperature refrigerator and freeze–thawed twice. After defrosting the plates, the contents of the neutralized plates were transferred to 96-well plates that contained the rotavirus antigen. The plates were then incubated at 37 °C for 1 h. The Institute of Medical Biology, Chinese Academy of Medical Sciences, produced HRP-labeled anti-rotavirus antibodies and added 100 µL/well at a dilution ratio of 1:2000. Plaques were kept at 37 °C for 1 h to facilitate binding. After each of the five washes in the plate washer with the PBST solution, the plates were tapped dry with filter paper. For reliable findings after color development, the ELISA plate was tapped dry before proceeding. Color development was achieved by adding 100 µL of TMB (3,3′,5,5′-tetramethylbenzidine) to each well in a darkened chamber at ambient temperature and incubating for 2–10 min. To stop color development, 100 µL of 2M H_2_SO_4_ was added to each well. A reference wavelength of 650 nm and an absorbance value of 450 nm were determined using an enzyme spectrophotometer. The number that was used to determine the cutoff was arrived at by multiplying the mean of the control cell assay by 2.1. A rotavirus-neutralizing-antibody-potency-positive well was defined as a serum test result in which the detection value was lower than the computed threshold. 

### 2.4. Determination of IgG and IgA Antibody Titers

The Institute of Medical Biology, Chinese Academy of Medical Sciences, produced the rotavirus antigen ELISA 96-well plates. Each serum dilution required the preparation of two wells, and after that, the plates underwent incubation at 37 degrees Celsius for a single hour. After being cleaned five times with PBST solution, the plates were blotted dry using filter paper. HRP-labeled anti-human IgG and HRP-labeled anti-human IgA were both added to the sample at 100 µL/well for the purpose of determining IgG and IgA, respectively, with a dilution factor of 1:12,000. After that, the plates were incubated at 37 °C for one hour. After being cleaned five times with PBST solution in a plate washer, the plates were blotted dry using filter paper. The ELISA plates were incubated in a dark chamber at room temperature for 2–10 min after 100 µL of TMB was added to each well for color development. To stop color development, 100 µL of 2M H_2_SO_4_ was added to every well. A reference wavelength of 650 nm and an absorbance value of 450 nm were determined using an enzyme spectrophotometer.

### 2.5. Data Selection for Pre-Blinding Analysis

As described by the intention-to-treat (ITT) principle, the full analysis set (FAS) depicts the ideal participant population. All individuals who fulfilled the inclusion/exclusion criteria were randomly assigned to receive the experimental vaccine, and those who had past, present, or future results from any blood tests conducted prior to, during, or after the final immunization were included in the FAS. At that time, the FAS included these people to assess immunogenicity. The safety analysis set (SS) comprised all subjects who were administered the experimental vaccination after randomization and for whom data from a minimum of one safety assessment was accessible. 

### 2.6. Analytical Methods

The following section presents the demographic baseline characteristics of the study population. To establish balance, an χ^2^ test was employed to ascertain whether there were any significant differences in the distribution of sexes among participants. The adherence analysis examined the extent to which participants adhered to the prescribed regimen. The study described the enrollment, dropout, and exclusion rates, as well as the number of cases with blood collection; the reasons for dropping out; and the medication and vaccine combinations in the test and placebo groups. To ascertain whether there were any significant differences in dropout and combination rates between the test and placebo groups across all age categories, an χ^2^ test, a corrected χ^2^ test, or the exact probability method were employed.

In order to assess the possible dangers of the suggested layout, a safety study was carried out. We used the SS as the basis for our safety study. We calculated the incidence rate and 95% CI of adverse reactions by counting the number of events and responses in both the test and placebo groups. The constitutive ratio was used to indicate the severity of the adverse effects. To determine whether the frequencies of adverse events (AEs), Grade 3 AEs, and serious adverse events (SAEs) were significantly different between the test group and the placebo group for each age group, the exact probability technique, the corrected χ^2^ test, and the χ^2^ test were all used. We used a rank-sum test to see whether there was a significant difference in the mean adverse reaction grades between the test and placebo groups for each age group of individuals who had been adversely affected.

The results of the immunogenicity test were analyzed. GMT, and 95% CI were used to describe the data, which were derived from logarithmic transformations of the antibody titers. The treatment groups were compared based on the pre-immunization serum antibody GMT. In order to compare the experimental and placebo groups, the antibody titers were log-transformed. Two independent sample t-tests or corrected t-tests were used for comparisons where the data were normal but had heteroskedastic variance. There was no significant difference in the pre-immunization serum anti-rotavirus neutralizing antibody, ELISA antibody, or serum IgA antibody GMTs among the placebo group subjects. The log-transformed antibody titers were compared at post-immunization timepoints using two independent sample t-tests or corrected *t*-tests.

## 3. Results

### 3.1. Safety

The safety analysis comprised 96 teenagers (ranging in age from 6 to 17 years) who were part of the adolescent group and had data from a minimum of one safety assessment (Appendix A). There were 66 adverse events (AEs) documented between 0 and 30 days after complete immunization, with 22 of them being vaccine-related. A statistically significant difference was observed in the incidence of total AEs within 0 to 30 days following final vaccination between the low-dose group and the placebo group. No significant difference was noted in the other groups. A statistically significant difference was observed in the incidence of total adverse reactions (Grade 1) within 30 days after final vaccination between the low-dose and placebo groups. Appendix A show that between 0 and 30 days after final vaccination, the placebo group and the groups given different dosages did not differ significantly in terms of the incidence or severity of adverse events, adverse reactions, symptomatic adverse reactions, or total adverse reactions, regardless of whether the adverse event was local, systemic, or otherwise. Cough, fever, lethargy, redness at the injection site, soreness at the injection site, swelling at the injection site, constipation, diarrhea, and abnormal laboratory indicators of adverse events (AEs) were seen between 0 and 30 days after complete immunization. When compared to the placebo group, there was no statistically significant difference in the occurrence of related symptoms across the dosing groups (Appendix A). There was a preponderance of Grade 1 and 2 adverse events in the experimental group. We did not register any SAEs. Two serious adverse events (AEs) of Grade 3 or higher that were determined to be unrelated to the vaccine were documented in the 6–17-year-old group after the whole course of immunization. One case occurred between 0 and 30 days after the initial vaccination dose. The subject was in the high-dose group, and the symptom was the presence of protein in the urine. The second case occurred between 0 and 30 days after the second dose of the vaccine. The subject was in the placebo group and exhibited symptoms indicative of an abnormal white blood cell count (Appendix A). The incidence of Grade 3 AEs related to the trial vaccine was zero (Table 1 and Table 2).

Local adverse reactions observed in the adolescent group following test vaccine administration included vaccination site erythema, site of vaccination pain, and vaccination site swelling. Systemic adverse reactions included pyrexia, cough, fatigue, constipation, and diarrhea. The majority of adverse reactions were Grade 1, with a few Grade 2 reactions (Table 3).

### 3.2. Immunogenicity

The neutralizing antibody potency assay showed a dose-dependent effect for both the neutralizing antibody GMT and geometric mean increase (GMI) 28 days after final immunization, with GMT (95% CI) values of 424.32 (283.05–636.10), 504.63 (358.79–709.76), and 925.45 (708.10–209.52) and GMI (95% CI) values of 1.86 (1.24–2.79), 2.15 (1.45–3.18), and 3.57 (2.84–4.48) for the low-, medium-, and high-dose treatment groups, respectively. The seroconversion rate of the neutralizing antibody after final immunization was 0.00% (0.00 to 14.25) in the placebo group, 8.33% (1.03 to 27.00) in the low-dose group, 33.33% (15.63 to 55.32) in the medium-dose group, and 54.17 (32.82 to 74.45) in the high-dose group. Based on these findings, it seems that the high-dose group was the only one whose immunization resulted in positive neutralizing antibodies, and the production of neutralizing antibodies demonstrated a dose-related effect (the R^2^ value was 0.9737) (Figure 1, Appendix A).

The GMT (95% CI) values of serum rotavirus-specific IgG at 28 days after final immunization were 6501.86 (4047.23–10,445.2), 6501.82 (3718.35–11,368.9), and 10,173.3 (4512.65–22,934.8), with GMI (95% CI) values of 6.44 (3.64–11.40), 6.82 (4.40–10.57), and 11.99 (5.73–25.09) in the low-, medium-, and high-dose groups, respectively. The GMT in serum rotavirus-specific IgG showed an almost twofold increase in the high-dose group relative to the low- and medium-dose groups, and the GMI showed a dose correlation. The seroconversion rate (95% CI) of IgG at 28 days after final immunization was 4.17 (0.11–21.12), 62.50 (40.59–81.20), 79.17 (57.85–92.87), and 91.67 (73.00–98.97) in the placebo, low-dose, medium-dose, and high-dose groups, respectively, indicating that only the high-dose group produced an IgG antibody seroconversion rate higher than 90% (Figure 2, Appendix A). The production of IgG antibodies demonstrated a dose-related effect (the R^2^ value was 0.8468).

The GMT (95% CI) values of serum rotavirus-specific IgA at 28 days after final immunization were 2733.64 (1465.57–5098.91), 2233.29 (1231.49–4050.03), and 3596.66 (1957.20–6609.41), with GMI (95% CI) values of 4.18 (2.37–7.37), 5.34 (2.71–10.53), and 9.93 (5.49–17.97) for the low-, medium-, and high-dose treatment groups, respectively. The GMI of serum-specific IgA showed a dose-related correlation. After 28 days of complete vaccination, the placebo group had an IgA seroconversion rate of 0.00% (0.00–14.25), the low-dose group had one of 70.83% (48.91–87.38), and the high-dose group had one of 79.17% (57.85–92.87) (Figure 3, Appendix A).

## 4. Discussion

Based on our results, it seems that adolescents have no problems with inactivated rotavirus vaccination. No significant adverse events (SAEs) were reported, and most local and systemic responses were moderate and short-lived after immunization. This safety profile is comparable to that of other widely used vaccines [30,31,32], providing substantial support for the clinical promotion of this vaccine. Statistical analysis revealed that the low-dose group had a significantly greater incidence of overall adverse responses compared to the placebo, medium-dose, and high-dose groups. On the other hand, as compared to the placebo group, neither the medium-dose nor the high-dose group showed a statistically significant increase or decrease in the incidence of overall adverse events.

No statistically significant differences were observed in the incidence of localized and systemic adverse reactions between the vaccine (when all three doses were combined) and placebo groups. The occurrence of vaccination site erythema, swelling, and pain may be due to the adjuvant components present in the vaccine, the injection procedure, and/or the inherent variability in the immune responses of individuals. The incidence and severity of local reactions can be further reduced by optimizing the vaccine’s formulation, improving the injection technique, and strengthening local care after vaccination. Systemic reactions such as pyrexia and fatigue may be caused by the vaccine stimulating the immune system to produce an immune response and release inflammatory mediators. In the case of systemic reactions such as mild fever, the use of symptomatic measures such as physical cooling and increased fluid intake may be beneficial. There was no statistically significant difference between the dosage groups and the placebo group in terms of symptomatic side events, such as cough, constipation, or diarrhea, from 0 to 30 days after immunization. In a previous study, the safety and immunogenicity of different doses of the vaccine were systematically evaluated in a population of healthy adults. This novel inactivated rotavirus vaccine was generally well tolerated in adults, and the vaccine was immunogenic in adults [28]. A synthesis of the results from safety studies of the inactivated rotavirus vaccine in both adult and adolescent populations indicates that the vaccine demonstrates a favorable safety profile.

It was first shown that in subjects aged 6–17 years, there was a dose-related effect, with higher neutralizing antibody levels in the high-dose group compared to the medium- or low-dose groups, when considering the GMT and positive seroconversion rate of neutralizing antibodies for the primary immunogenicity endpoints in the adolescent group. The potency of neutralizing antibodies, IgG antibodies, and IgA antibodies exhibited an increasing pattern of GMT and seroconversion rate with increasing immunization dose. According to available reports, all currently available rotavirus vaccines have demonstrated good immunogenicity [30,33,34,35,36]. Currently, three rotavirus vaccines are available in China, all of which are oral, live, and attenuated vaccines. There is considerable scope for improvement in terms of vaccine types, strategies, and components, as well as protective effects. The inactivated rotavirus vaccine has the potential to be less prone to virulence regression and infection than the live attenuated rotavirus vaccine [37,38,39]. However, its immunogenicity may be relatively weak. Consequently, further research is required to optimize the vaccine’s immunization strategy. This may entail increasing the vaccine dose, adjusting the number of vaccinations, or combinations with immune enhancers. The aim is to enhance the vaccine’s immunity effect while maintaining a favorable safety profile.

The current immunization strategy for rotavirus vaccines is primarily focused on infants and young children, and the most optimal vaccination strategy for the adolescent population remains undetermined. For instance, future studies need to fine the optimal vaccination dose, the number of vaccinations, and the intervals between vaccinations and other vaccines. As a result, it is challenging to establish standardization in the administration of vaccines to the adolescent population. Additionally, the cost of the rotavirus vaccine may present a financial barrier to vaccination for some adolescents from economically disadvantaged families. While some regions have implemented vaccination subsidy programs, there are still adolescents who cannot benefit from them, which may limit the widespread uptake of the vaccine in the adolescent population. While this study offers a comprehensive evaluation of the clinical safety of the inactivated rotavirus vaccine in an adolescent population, it is not without limitations. For instance, the observational period of this study was relatively brief, and the long-term safety concerns (e.g., potential risk of immune-related diseases) of the vaccine remain unresolved. Additionally, the study samples were primarily drawn from specific regions and populations in China, which may introduce geographical limitations. Larger-scale, longer-term, multi-regional, and multi-ethnic phase II and III clinical trials must be conducted to further study the safety and efficacy of the inactivated rotavirus vaccine. This will facilitate safer and more effective means of rotavirus prophylaxis for infants and young children.

## 5. Conclusions

Regarding safety, the majority of adverse events were Grade 1 or 2, suggesting that the experimental vaccine is safe for healthy adolescents. There was a dose-dependent increase in neutralizing antibody and IgG antibody production after inactivated rotavirus vaccine injection, with antibody GMT titers and the GMI in antibody levels rising in proportion to the administered dose.

## Figures and Tables

**Figure 1 vaccines-13-00369-f001:**
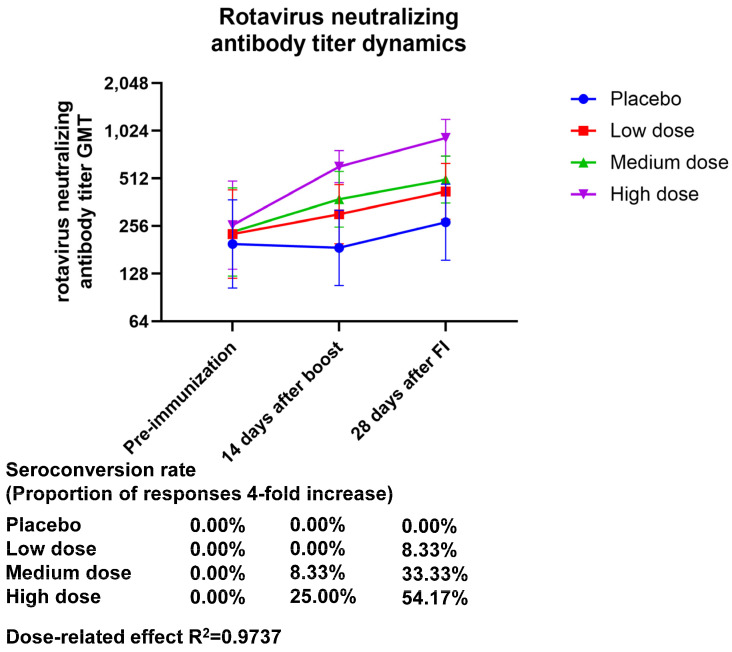
Dynamics of the neutralizing antibody titers of anti-rotavirus generated by vaccination. Zero days after the first injection was the pre-immunization timepoint, fourteen days following the booster shot was forty-two days following the initial injection, and twenty-eight days following the complete shot was fifty-six days following the initial injection.

**Figure 2 vaccines-13-00369-f002:**
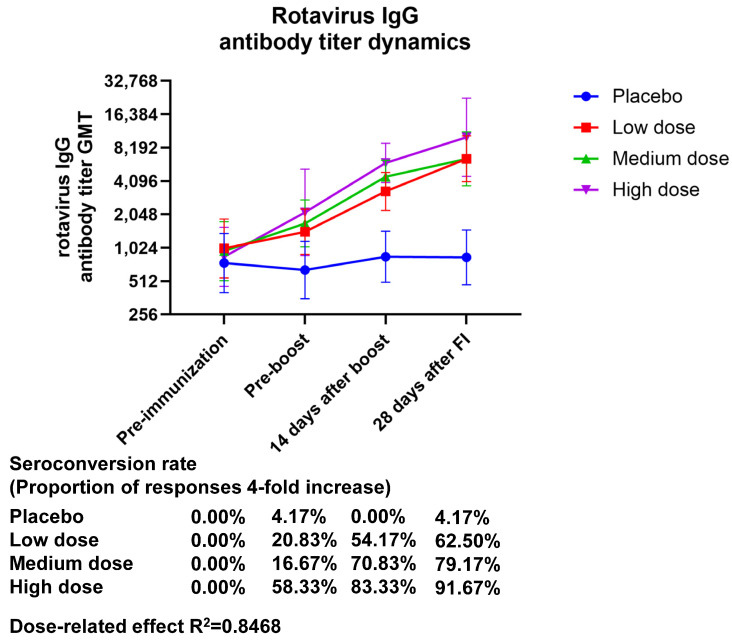
Changes in anti-rotavirus IgG antibody titers brought about by vaccination over time. Zero days after the first injection was the pre-immunization timepoint, fourteen days following the booster shot was forty-two days following the initial injection, and twenty-eight days following the complete shot was fifty-six days following the initial injection.

**Figure 3 vaccines-13-00369-f003:**
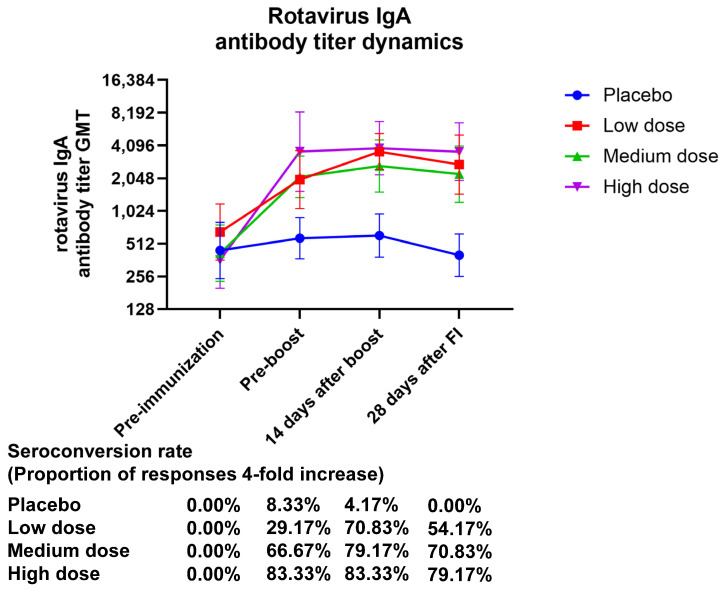
The temporal pattern of anti-rotavirus IgA antibody titers elicited by vaccination. Zero days after the first injection was the pre-immunization timepoint, fourteen days following the booster shot was forty-two days following the initial injection, and twenty-eight days following the complete shot was fifty-six days following the initial injection.

**Table 1 vaccines-13-00369-t001:** Incidence of adverse events sorted by type 0–30 days after final vaccination in the different treatment groups.

	Placebo (N = 24)		Low Dose (N = 24)			Medium Dose (N = 24)			High Dose (N = 24)		
	N (%)	No. of Events	N (%)	*p*	No. of Events	N (%)	*p*	No. of Events	N (%)	*p*	No. of Events
Total adverse events	2 (8.33)	3	9 (37.50)	0.016	12	3 (12.50)	>0.999	5	1 (4.17)	>0.999	2
Local adverse events	1 (4.17)	1	6 (25.00)	0.102	6	1 (4.17)	>0.999	2	1 (4.17)	>0.999	2
Systemic adverse events	2 (8.33)	2	4 (16.67)	0.663	6	3 (12.50)	>0.999	3	0 (0.00)	0.470	0

*p* values are for comparisons between each treatment group and the placebo group.

**Table 2 vaccines-13-00369-t002:** Graded incidence of adverse reactions sorted by type 0–30 days after final vaccination in the different treatment groups.

		Placebo (N = 24)		Low Dose (N = 24)		Medium Dose (N = 24)		High Dose (N = 24)	
		N (%)	No. of Events	N (%)	No. of Events	N (%)	No. of Events	N (%)	No. of Events
Total adverse reactions	Grade 1	2(8.33)	3	8 (33.33)	10	3 (12.50)	4	1 (4.17)	2
	Grade 2	0 (0.00)	0	2 (8.33)	2	1 (4.17)	1	0 (0.00)	0
	Grade 3	0 (0.00)	0	0 (0.00)	0	0 (0.00)	0	0 (0.00)	0
Local adverse reactions	Grade 1	1(4.17)	1	6 (25.00)	6	1 (4.17)	1	1(4.17)	2
	Grade 2	0 (0.00)	0	0 (0.00)	0	1 (4.17)	1	0 (0.00)	0
	Grade 3	0 (0.00)	0	0 (0.00)	0	0 (0.00)	0	0 (0.00)	0
Systemic adverse reactions	Grade 1	2 (8.33)	2	3 (12.50)	4	3 (12.50)	3	0 (0.00)	0
	Grade 2	0 (0.00)	0	2 (8.33)	2	0 (0.00)	0	0 (0.00)	0
	Grade 3	0 (0.00)	0	0 (0.00)	0	0 (0.00)	0	0 (0.00)	0

**Table 3 vaccines-13-00369-t003:** Grade analysis of adverse reactions to final immunization with the different doses of the test vaccine and placebo in the adolescent group.

Intensity	Group (N = 24)	Grade 1 N (%)	Grade 2 N (%)	Grade 3 N (%)
Respiratory, thoracic, and mediastinal disorders				
Cough	Placebo	1 (4.17)	0 (0)	0 (0)
	Low dose	1 (4.17)	0 (0)	0 (0)
	Medium dose	0 (0)	0 (0)	0 (0)
	High dose	0 (0)	0 (0)	0 (0)
General disorders and administration site conditions				
Pyrexia	Placebo	0 (0)	0 (0)	0 (0)
	Low dose	1 (4.17)	2 (8.33)	0 (0)
	Medium dose	1 (4.17)	0 (0)	0 (0)
	High dose	0 (0)	0 (0)	0 (0)
Fatigue	Placebo	0 (0)	0 (0)	0 (0)
	Low dose	0 (0)	0 (0)	0 (0)
	Medium dose	1 (4.17)	0 (0)	0 (0)
	High dose	0 (0)	0 (0)	0 (0)
Vaccination site erythema	Placebo	0 (0)	0 (0)	0 (0)
	Low dose	0 (0)	0 (0)	0 (0)
	Medium dose	0 (0)	1 (4.17)	0 (0)
	High dose	0 (0)	0 (0)	0 (0)
Site of vaccination painful	Placebo	1 (4.17)	0 (0)	0 (0)
	Low dose	6 (25.00)	0 (0)	0 (0)
	Medium dose	0 (0)	0 (0)	0 (0)
	High dose	1 (4.17)	0 (0)	0 (0)
Vaccination site swelling	Placebo	0 (0)	0 (0)	0 (0)
	Low dose	1 (4.17)	0 (0)	0 (0)
	Medium dose	0 (0)	0 (0)	0 (0)
	High dose	0 (0)	0 (0)	0 (0)
Gastrointestinal disorders				
Constipation	Placebo	0 (0)	0 (0)	0 (0)
	Low dose	1 (4.17)	0 (0)	0 (0)
	Medium dose	0 (0)	0 (0)	0 (0)
	High dose	0 (0)	0 (0)	0 (0)
Diarrhea	Placebo	1 (4.17)	0 (0)	0 (0)
	Low dose	1 (4.17)	0 (0)	0 (0)
	Medium dose	1 (4.17)	0 (0)	0 (0)
	High dose	0 (0)	0 (0)	0 (0)

## Data Availability

Data-sharing inquiries should be directed to the authors.

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
