# Peer review of "Safety and Immunogenicity of a New Rotavirus-Inactivated Vaccine in the Chinese Adolescent Population: A Randomized, Double-Blind, Placebo-Controlled Phase I Clinical Trial"

_vaccines, 2025, doi:10.3390/vaccines13040369_

Round 1
Reviewer 1 Report
Comments and Suggestions for Authors
The article entitled “Safety and immunogenicity of a new rotavirus-inactivated vaccine in the Chinese adolescent population: A randomized, double-blind, placebo-controlled phase I clinical trial” is self-explanatory in nature. A vaccine against rotavirus has been developed and a phase I trial has been accomplished to proceed with its usage in future. The vaccines seem to be safe of acceptable level and immunogenicity is of considerable grade. As the authors mentioned, there are several vaccines against rotavirus in the world, including China. These vaccines are also of considerable safety and potent immunogenic.
Comments
- The authors have mentioned that they have produced a new rotavirus-inactivated vaccine. What is the new part of the vaccine? Is the usage of the vaccine in adolescents?
- If that is the case, several other rotavirus vaccines that are found in markets and used in infants or children are ineffective in adolescents?
- The article can be ideally having a group of controls that received commercially available vaccines for the infants and children.
- The antibody titers of vaccines in various doses have been shown. However, it would be clinically relevant to assess if the vaccine for infants also provides equal, better, or lower antibodies in adolescents.
Author Response
For research article
|
Response to Reviewer 1 Comments
|
- Summary
Reviewer comment: The article entitled “Safety and immunogenicity of a new rotavirus-inactivated vaccine in the Chinese adolescent population: A randomized, double-blind, placebo-controlled phase I clinical trial” is self-explanatory in nature. A vaccine against rotavirus has been developed and a phase I trial has been accomplished to proceed with its usage in future. The vaccines seem to be safe of acceptable level and immunogenicity is of considerable grade. As the authors mentioned, there are several vaccines against rotavirus in the world, including China. These vaccines are also of considerable safety and potent immunogenic.
Response to the comment:
We would like to thank reviewer 1 for the constructive and insightful advice. We have addressed all the points raised by the reviewer, as summarized below.
In view of the high morbidity and mortality associated with rotavirus, the World Health Organization recommends rotavirus vaccination to prevent rotavirus diarrhea. The use of live attenuated vaccines for the prevention of viral diseases involves certain risks, and there is an urgent need for a new type of vaccine that is more safe, effective, of controlled quality, and easy to use for immunization. The progress of science and technology and the attention paid to rotavirus have greatly facilitated the development of rotavirus vaccines, and different types of new vaccines have begun to appear. Among them, non-replicating vaccines are considered to be the most promising.
Rotavirus vaccine is indicated for infants and children under 5 years of age. Our inactivated rotavirus vaccine also targets children under 5 years of age. Inactivated rotavirus vaccine has not been licensed in the world. Although the target population of this vaccine is infants and young children, phase I clinical trials need to start with adults, then in adolescent population for safety reasons. In Introduction section (line 85-89), we have added the following statement: The target population for inactivated rotavirus vaccine is infants and children under the age of 5 years. Safety and immunogenicity were evaluated in the adolescent population to evaluate the safety of different doses of the vaccine and to provide a database for the subsequent design of the immunization program.
|
2. Questions for General Evaluation |
Reviewer’s Evaluation |
Response and Revisions |
|
Does the introduction provide sufficient background and include all relevant references? |
Must be improved |
Agree, we have, accordingly revised the manuscript. |
|
Is the research design appropriate? |
Must be improved |
|
|
Are the methods adequately described? |
Can be improved |
Agree, we have, accordingly revised the manuscript. |
|
Are the results clearly presented? |
Must be improved |
Agree, we have, accordingly revised the manuscript. |
|
Are the conclusions supported by the results? |
Must be improved |
|
- Point-by-point response to Comments and Suggestions for Authors
Reviewer comment 1: The authors have mentioned that they have produced a new rotavirus-inactivated vaccine. What is the new part of the vaccine? Is the usage of the vaccine in adolescents?
Response to the comment 1:
Many thanks for your expert opinion. Inactivated rotavirus vaccine has not been licensed in the world. A novel preparation process was employed to manufacture a new non-replicating inactivated rotavirus vaccine. In this study, inactivated rotavirus vaccine using a human wild-type rotavirus named ZTR-68-A (G1P[8]) isolated from a child's stool with diarrhoea in China. ZTR-68-A ( genotype G1-P[8]-I1-R1-C1-M1-A1-N1-T1-E1-H1) was manufactured by the Institute of Medical Biology, Chinese Academy of Medical Science. The virus was cultivated in Vero cells at an MOI of 0.1 in serum-free MEM medium, clarified by centrifugation, and then concentrated by ultrafiltration. Virus purification was performed using ion exchange chromatography and molecular sieve chromatography. Virus was inactivated with formaldehyde, and produced by adding aluminum hydroxide adjuvant, was assessed the safety and immunogenicity in Chinese adolescent population. Preclinical studies have shown the inactivated rotavirus vaccine can induce serum neutralizing antibodies and provide protection in animal models.
The virus strain in this study is a new strain of human origin and is isolated and prepared as an inactivated vaccine using a self-developed purification and inactivation process. We have added a paragraph to describe the summary of the development of this inactivated rotavirus vaccine in the supplementary data.
The inactivated rotavirus vaccine is not intended for use in adolescents. Rotavirus vaccine is indicated for infants and children under 5 years of age. Our inactivated rotavirus vaccine also targets children under 5 years of age. Inactivated rotavirus vaccine has not been licensed in the world. Although the target population of this vaccine is infants and young children, phase I clinical trials need to start with adults, then in adolescent population for safety reasons.
Reviewer comment 2: If that is the case, several other rotavirus vaccines that are found in markets and used in infants or children are ineffective in adolescents?
Response to the comment 2:
Thank you for your question. The target population for currently available rotavirus vaccines is infants and young children. Vaccines intended for use in different target populations should undergo independent clinical trials and be evaluated separately for efficacy and safety. As far as the clinical study data we queried, there are no existing marketed rotavirus vaccines that have been evaluated for vaccine efficacy in the adolescent population. It is therefore difficult to assess whether marketed vaccines are effective or ineffective in the adolescent population.
Reviewer comment 3: The article can be ideally having a group of controls that received commercially available vaccines for the infants and children.
Response to the comment 3:
The objective of the present study was to evaluate the safety and immunogenicity of IRV in infants and children. To date, no non-replicating vaccine has received approval, and no commercial vaccine has been developed that utilizes the same technology platform. Your suggestion is noteworthy, and in subsequent research endeavors, we will establish a live attenuated vaccine control group that is appropriate for infants and young children to facilitate a comparative analysis of rotavirus vaccines based on different technological routes (inactivated vaccine and attenuated live vaccine).
Reviewer comment 4: The antibody titers of vaccines in various doses have been shown. However, it would be clinically relevant to assess if the vaccine for infants also provides equal, better, or lower antibodies in adolescents.
Response to the comment 4:
Thank you very much for your advice! This study focuses on the safety and preliminary immunogenicity of an inactivated rotavirus vaccine in a population of adolescents aged 6-17 years. The results of the work presented in this paper are the results of a clinical phase I a trial. The data were not presented in this article, we will publish data from clinical trials in other age including 2-6 month, and 7-71 month groups separately.

Reviewer 2 Report
Comments and Suggestions for Authors
The authors report an important clinical phase I trial towards a new rotavirus-inactivated vaccine. The manuscript fits the scope of the journal, and it requires some revision before its acceptance for publication.
General comments:
I cannot see a definition and/or indicated reference as concerns AE and SAE, ca. lines 119-211. From the text and tables 2 and 3 it appears that AE span from Grade 1 to Grade 3, whereas SAE would be Grade 4, is this correct?
I am not sure about the exact study design described in lines 111 to ca. 140, in Figures 1-3 and outlines in Figure S1. Can the authors please confirm or correct my following interpretations? Pre-immunization means the levels (neutralizing antibody titer, IgG and IgA titres) before the campaign started. Day 0 is the day of the first injection. 14 days later, samples are taken and analysed. At (or very soon after this time point), the second injection was applied. 28 days after the second injection, again samples taken and analysed.
Various Tables and Supplementary Tables: how are number of events and number of samples defined? What is the difference between Precedent and number of examples? How is a statistical correlation distinguished from “irrelevant”?
Specific comments:
Abstract
Line 20, placebo-controlled
Materials and Methods
Line 93, vaccine?
Line 95, a reference and/or definition should be added for EU
Line 104, risks
Lines 112-114, I am not sure about the meaning of this sentence, first, the authors say that it was checked whether Grade 3 responses were identified at all, and then they say “ whether such reactions did not surpass 15 %”, why 15%?
Lines 120-121, SAEs observed after a 6 months monitoring period? This is the only time in the manuscript relating to SAEs, afterwards this is not picked out as a central theme. I guess the sentence is supposed to read “SAEs were only observed during …”/
Line 145, the wells were filled
Line 155, defrosted? Do you mean that the plates cooled down from 37 degrees to ambient temperature?
Line 162, the ELISA plate was tapped dry
Lines 165 and 181, reference wavelength
Lines 166 and 182, spectrophotometer
Section 2.5, the last sentence is not clear; why defining rules and nevertheless allow participation if the rules were not (fully) respected?
Results
Lines 230 and 231, where can the 66 and 22 cases be seen in Table S1?
Lines 232, 243 and several more times in the text, 0-28 days, in the Tables, 0-30 days are mentioned
Table 1, shouldn’t the numbers for the total adverse events (sum of local and systemic events) be 3, 10, 4 and 1 instead of 2, 9 3 and 1?
Line 282, where can the r2 values be seen in Table S6 and/or Figure 1?
Tables 1-3 and Figures S6-S8; the biggest effect lies in the increase of the neutralizing antibody GMT, the vaccination effect is less pronounces in the IgG-specific levels (no different between low and medium dose, high dose only 50 % higher than these) and the IgA-specific levels (even a lower titer for the medium dose compared to the low dose; can this be explained?)
Discussion
Lines 328-330, how can the described effect be explained?
Line 367, find?
I fully agree with the authors as regards additional studies, e.g. monitoring of long-term effects, minimizing adverse effects from adjuvants, and especially larger-scale studies with hundreds of participants rather than 24 per group.
Figure S1 and general comments above: maybe include a graph which shows the vaccination scheme and when samples for analysis were taken to make it clearer.
Author Response
For research article
|
Response to Reviewer 2 Comments
|
- Summary
Reviewer comment: The authors report an important clinical phase I trial towards a new rotavirus-inactivated vaccine. The manuscript fits the scope of the journal, and it requires some revision before its acceptance for publication.
Response to the comment:
We thank the Reviewer 2 for the careful read and thoughtful comments on previous draft. The comments are all valuable and were very helpful in revising and improving our paper, as well as for providing important guidance regarding our research.
|
2. Questions for General Evaluation |
Reviewer’s Evaluation |
Response and Revisions |
|
Does the introduction provide sufficient background and include all relevant references? |
Yes |
|
|
Is the research design appropriate? |
Can be improved |
|
|
Are the methods adequately described? |
Can be improved |
Agree, we have, accordingly revised the manuscript. |
|
Are the results clearly presented? |
Can be improved |
Agree, we have, accordingly revised the manuscript. |
|
Are the conclusions supported by the results? |
Can be improved |
|
- Point-by-point response to Comments and Suggestions for Authors
Reviewer comment 1: I cannot see a definition and/or indicated reference as concerns AE and SAE, ca. lines 119-211. From the text and tables 2 and 3 it appears that AE span from Grade 1 to Grade 3, whereas SAE would be Grade 4, is this correct?
Response to the comment 1:
Many thanks for your careful review. An AE is any untoward medical occurrence in a patient or clinical investigation subject administered a pharmaceutical product and that does not necessarily have a causal relationship with this treatment. Serious Adverse Event (SAE) or Serious Adverse Drug Reaction (Serious ADR), any untoward medical occurrence that at any dose: results in death, is life-threatening, requires inpatient hospitalization or prolongation of existing hospitalization, results in persistent or significant disability/incapacity, or is a congenital anomaly/birth defect. (International Conference on Harmonization of Technical Requirements for Registration of Pharmaceuticals for Human Use - Good Clinical Practice, ICH Official web site : ICH)
Grade 1 is mild, Grade 2 is moderate, Grade 3 is severe or medically significant, Grade 4 is life-threatening and requires urgent treatment, and Grade 5 is death. Comparing the SAE definitions, Grades 4 and 5 meet the SAE criteria.
Reviewer comment 2: I am not sure about the exact study design described in lines 111 to ca. 140, in Figures 1-3 and outlines in Figure S1. Can the authors please confirm or correct my following interpretations? Pre-immunization means the levels (neutralizing antibody titer, IgG and IgA titres) before the campaign started. Day 0 is the day of the first injection. 14 days later, samples are taken and analysed. At (or very soon after this time point), the second injection was applied. 28 days after the second injection, again samples taken and analysed.
Response to the comment 2:
Many thanks for the insightful comments.
Pre-immunization means the levels (neutralizing antibody titer, IgG and IgA titres) before the campaign started. That is correct.
Day 0 is the day of the first injection. 14 days later, samples are taken and analysed. That is correct.
The second injection was applied at 28 days after first injection. 28 days after the second injection (final injection), again samples taken and analysed.
Reviewer comment 3: Various Tables and Supplementary Tables: how are number of events and number of samples defined? What is the difference between Precedent and number of examples? How is a statistical correlation distinguished from “irrelevant”?
Response to the comment 3:
Thank you very much for pointing this out. The vocabulary of the number of occurrences and number of subjects in the table is inaccurate. Number of occurrences: An occurrence usually refers to the number of times an AE occurs. Number of subjects: number of subjects presenting AE reports. We have checked our statistical data and amended tables.
The vocabulary of statistical correlation and irrelevant is inaccurate. We have amended tables with “correlation” and “irrelevance”.
Reviewer comment 4: Line 20, placebo-controlled
Response to the comment 4:
Thank you for your revision!
Reviewer comment 5: Line 93, vaccine?
Response to the comment 5:
Thank you for your revision!
Reviewer comment 6: Line 95, a reference and/or definition should be added for EU
Response to the comment 6:
We established an antigenic content assay protocol based on the ELISA antigenic activity of the vaccine and applied it to the confirmation of the dose of this vaccine. Rotavirus vaccine antigen detection methods we put in the supplementary material in No.5 page, “S8. Determination of inactivated rotavirus vaccine antigen content”.
Reviewer comment 7: Line 104, risks
Response to the comment 7:
Thank you for your revision!
Reviewer comment 8: Lines 112-114, I am not sure about the meaning of this sentence, first, the authors say that it was checked whether Grade 3 responses were identified at all, and then they say “ whether such reactions did not surpass 15 %”, why 15%?
Response to the comment 8:
Thank you very much for this question. In the safety assessment on day 7 after the first vaccine dose, the results of the preliminary safety assessment were reviewed to determine whether the requirements of the program were met, i.e., there are no cases of vaccine-related Grade 4 serious adverse reactions or serious unexpected suspected adverse reactions, or the number of participants with adverse reactions of Grade 3 severeness does not exceed 15% of vaccinated participants in the group. No more than 15% is the safety standard we set in the clinical trial protocol. In the event that the number of subjects experiencing adverse reactions of Grade 3 exceeds 15% of the subjects in the cohort, the formation of an independent Data Safety and Monitoring Board becomes mandatory. The committee's primary responsibility is to validate safety and determine whether to suspend or terminate the trial.
Reviewer comment 9: Lines 120-121, SAEs observed after a 6 months monitoring period? This is the only time in the manuscript relating to SAEs, afterwards this is not picked out as a central theme. I guess the sentence is supposed to read “SAEs were only observed during …”/
Response to the comment 9:
Thanks for pointing out the ambiguous writing. Lines 120-121 have rewritten as “All participants continued to be observed for adverse events from 8 to 30 days and records were kept. Participants were required to continue follow-up up to 6 months after full vaccination, with only serious adverse events (SAEs) recorded.”
Reviewer comment 10: Line 145, the wells were filled
Response to the comment 10:
Thank you for your revision!
Reviewer comment 11: Line 155, defrosted? Do you mean that the plates cooled down from 37 degrees to ambient temperature?
Response to the comment 11:
Thanks for pointing out the ambiguous writing. A description of one step of the test operation is missing here. “After incubation, plates were placed in a low-temperature refrigerator and freeze-thawed twice. After defrosting the plates, the contents of the neutralized plates were transferred to 96-well plates that contained the rotavirus antigen.”
Reviewer comment 12: Line 162, the ELISA plate was tapped dry
Response to the comment 12:
Thank you for your revision!
Reviewer comment 13: Lines 165 and 181, reference wavelength
Response to the comment 13:
Thank you for your revision!
Reviewer comment 14: Lines 166 and 182, spectrophotometer
Response to the comment 14:
Thank you for your revision!
Reviewer comment 15: Section 2.5, the last sentence is not clear; why defining rules and nevertheless allow participation if the rules were not (fully) respected?
Response to the comment 15:
Thanks for pointing out the ambiguous writing. This sentence conveys an inaccurate meaning and we have deleted it.
Reviewer comment 16: Lines 230 and 231, where can the 66 and 22 cases be seen in Table S1?
Response to the comment 16:
Thank you very much for this question. In Table S1, there were 66 adverse events (AEs) documented between 0 and 30 days after complete immunization. This included the number of 20 AEs occurred in the placebo group, 25 AEs occurred in the low-dose group, 12 AEs occurred in the medium-dose group and 9 AEs occurred in the high-dose group.
There were 22 adverse events (AEs) documented between 0 and 30 days after complete immunization being vaccine-related. This included the number of 3 AEs occurred in the placebo group, 12 AEs occurred in the low-dose group, 5 AEs occurred in the medium-dose group and 2 AEs occurred in the high-dose group.
Reviewer comment 17: Lines 232, 243 and several more times in the text, 0-28 days, in the Tables, 0-30 days are mentioned
Response to the comment 17:
Thank you very much for correcting my writing errors. The correct safety observation time is 0-30 days.
Reviewer comment 18: Table 1, shouldn’t the numbers for the total adverse events (sum of local and systemic events) be 3, 10, 4 and 1 instead of 2, 9 3 and 1?
Response to the comment 18:
Thank you very much for this question. The numbers shown in Table 1 were subject number (N) occurred in adverse events, where N=24 in each group. We added the missing event numbers in Table 1.
Reviewer comment 19: Line 282, where can the r2 values be seen in Table S6 and/or Figure 1?
Response to the comment 19:
Thank you very much for your professional advice! We have modified Figure 1 and 2, and added R2 values.
Reviewer comment 20: Tables 1-3 and Figures S6-S8; the biggest effect lies in the increase of the neutralizing antibody GMT, the vaccination effect is less pronounces in the IgG-specific levels (no different between low and medium dose, high dose only 50 % higher than these) and the IgA-specific levels (even a lower titer for the medium dose compared to the low dose; can this be explained?)
Response to the comment 20:
Thank you very much for this question.
We hypothesize that this may be related to the fact that rotavirus antibodies were already present in the adolescents and that levels of antibody were not consistent from one individual to another, which may be related to recent exposure of the individuals to the virus. Another possible reason is that inactivated vaccines may stimulate the body to produce serum antibody immune responses, such as IgG antibodies and neutralizing antibodies. Since our vaccine is an injectable inactivated vaccine, it primarily produces humoral IgG and neutralizing antibodies. For non-replicating inactivated vaccine immunogenicity studies, the focus is on neutralizing antibody potency. There is a dose correlation for this vaccine at the level of neutralizing antibodies.
Reviewer comment 21: Lines 328-330, how can the described effect be explained?
Response to the comment 21:
Thank you very much for this question. What we want to express in this section is that the vaccine has a good safety and adverse reaction tolerance.
Reviewer comment 22: Line 367, find?
Response to the comment 22:
Thank you for your revision!
Reviewer comment 23: I fully agree with the authors as regards additional studies, e.g. monitoring of long-term effects, minimizing adverse effects from adjuvants, and especially larger-scale studies with hundreds of participants rather than 24 per group.
Response to the comment 23:
Many thanks for your expert opinion and your professional advice. This study focuses on the safety and preliminary immunogenicity of an inactivated rotavirus vaccine in a population of adolescents aged 6-17 years. The results of the work presented in this paper are the results of a clinical phase I trial. We also tested the vaccine for immune long-term persistence and cross-neutralizing antibodies to rotavirus types in the target population of infants and young children (2-6 month age, and 7-71 month age). The data were not presented in this article. In Phase II clinical trials of the vaccine, we have also expanded the subject population and studied the vaccine dose immunogenicity and safety.
Reviewer comment 24: Figure S1 and general comments above: maybe include a graph which shows the vaccination scheme and when samples for analysis were taken to make it clearer.
Response to the comment 24:
Thank you very much for your professional advice! We have added a Figure S2 to show the vaccination scheme.

Round 2
Reviewer 1 Report
Comments and Suggestions for Authors
In the revised version of the article, the authors emphasized some points regarding the initial response of the reviewer:
- They mentioned that the vaccine is intended for infants and children below 5 years of age.
- They are the one group that is working with an inactivated Rota vaccine that may be most active and warranted at this moment.
- The inactivated Rotavirus vaccine has not been licensed in the world.
However, phase I clinical trial with the inactivated rotavirus vaccine has been accomplished already.
- A randomized, double-blind, placebo-controlled phase I clinical trial of rotavirus inactivated vaccine (Vero cell) in a healthy adult population aged 18-49 years to assess safety and preliminary observation of immunogenicity. Wu JY, Zhang W, Pu J, Liu Y, Huang LL, Zhou Y, Gao JM, Tan JB, Liu XL, Yang J, Lin XC, Feng GW, Yin N, Chen R, Hu XQ, Yi S, Ye J, Kuang XJ, Wang Y, Zhang GM, Sun MS, Wang YX, Hu ZY, Yang JS, Li HJ. Vaccine. 2024 Jul 25;42(19):4030-4039.
However, the authors have not referred to this article in their manuscript. This should be start line, and they can subsequently describe their vaccine.
I also recommended improving the research design, and this has not been responded to. Taken together, the novelty of the work is not of sufficient grade.
Author Response
For research article
|
Response to Reviewer 1 Comments (Round 2)
|
- Summary
Response to the comment:
Thank you very much for your expert insight. The inactivated rotavirus vaccine has not yet been completed for clinical trials and marketed for use. We have conducted various phases of Phase I clinical trials with inactivated rotavirus vaccines. The last study we completed was a safety and immunogenicity study of an inactivated rotavirus vaccine in a healthy adult population. It is also this reference that you mentioned. You have given us a very good suggestion to revise the article, as the lack of relevant description of the study objectives in this article could easily cause confusion with the published study of the safety and immunogenicity of inactivated rotavirus vaccine in a healthy adult population. We have added a discussion of the study objectives in the background section of the article as you suggested.
Phase I clinical trials are the first use of a new vaccine in humans, and one of the study components includes an initial safety and tolerability assessment in humans to explore recommended dosages and dosing regimens for subsequent clinical trials. Based on the vaccine's phase I clinical trial protocol, we used the age-climbing principle for safety reason to evaluate the safety and preliminary immunogenicity of the inactivated rotavirus vaccine in healthy populations of adults (18 to 49 years of age), adolescents (6 to 17 years of age), and infants and young children (2 months age to 6 months age, and 7 months age to 71 months age). Vaccine doses were also administered using a dose-climbing trial design to inform dose-exploratory studies for phase II clinical trials. Therefore, we designed the study with different age groups and different dose groups in the phase I clinical trial of the vaccine. In this study, we report a part of the work of the phase I clinical trial of the vaccine.
|
2. Questions for General Evaluation |
Reviewer’s Evaluation |
Response and Revisions |
|
Does the introduction provide sufficient background and include all relevant references? |
Must be improved |
|
|
Is the research design appropriate? |
Must be improved |
|
|
Are the methods adequately described? |
Must be improved |
|
|
Are the results clearly presented? |
Can be improved |
|
|
Are the conclusions supported by the results? |
Must be improved |
|
- Point-by-point response to Comments and Suggestions for Authors
Reviewer comment 1: However, phase I clinical trial with the inactivated rotavirus vaccine has been accomplished already. (A randomized, double-blind, placebo-controlled phase I clinical trial of rotavirus inactivated vaccine (Vero cell) in a healthy adult population aged 18-49 years to assess safety and preliminary observation of immunogenicity. Wu JY, Zhang W, Pu J, Liu Y, Huang LL, Zhou Y, Gao JM, Tan JB, Liu XL, Yang J, Lin XC, Feng GW, Yin N, Chen R, Hu XQ, Yi S, Ye J, Kuang XJ, Wang Y, Zhang GM, Sun MS, Wang YX, Hu ZY, Yang JS, Li HJ. Vaccine. 2024 Jul 25;42(19):4030-4039.)
Response to the comment 1:
Thank you for your suggestion. The inactivated rotavirus vaccine has not yet been completed for clinical trials and marketed for use. We have conducted various phases of Phase I clinical trials with inactivated rotavirus vaccines. The last study we completed was a safety and immunogenicity study of an inactivated rotavirus vaccine in a healthy adult population.
This study was also published by our research team. (A randomized, double-blind, placebo-controlled phase I clinical trial of rotavirus inactivated vaccine (Vero cell) in a healthy adult population aged 18-49 years to assess safety and preliminary observation of immunogenicity.)
In the discussion section of this thesis, we discuss the safety of an inactivated rotavirus vaccine in a phase I clinical trial in an adult population and find that the vaccine showed similar safety profiles in both adult and adolescent populations. We have added a relevant description in discussion section (line 359-364): In a previous study, the safety and immunogenicity of different doses of the vaccine were systematically evaluated in a population of healthy adults. This novel inactivated rotavirus vaccine was generally well-tolerated in adults, and the vaccine was immunogenic in adults. A synthesis of the results from safety studies of the inactivated rotavirus vaccine in both adult and adolescent populations indicates that the vaccine demonstrates a favorable safety profile.
Reviewer comment 2: However, the authors have not referred to this article in their manuscript. This should be start line, and they can subsequently describe their vaccine.
Response to the comment 2:
Thank you very much for your professional advice. We have added a description of the study objectives in the background section of the article as you suggested.
In the introductory section of the article, we have added a relevant description of the current and trial design of this clinical trial (line 84-90): The target population for inactivated rotavirus vaccine is infants and children under the age of 5 years. Phase I clinical trials of inactivated rotavirus vaccine have included adults, adolescents, infants and children. In a previous study, we systematically evaluated the safety and immunogenicity of different doses of the vaccine in a population of healthy adults (The corresponding reference is attached). Safety and immunogenicity were evaluated in the adolescent population to evaluate the safety of different doses of the vaccine and to provide a data-base for the subsequent design of the immunization program.
Reviewer comment 3: I also recommended improving the research design, and this has not been responded to. Taken together, the novelty of the work is not of sufficient grade.
Response to the comment 3:
Thank you for your professional suggestion.
Since Phase I clinical trials of inactivated rotavirus vaccines include safety and immunogenicity studies in different age groups and with different vaccine doses. The research effort and trial results are very large. The adolescent cohort group covered in this study was one of the age subgroups that were part of the Phase I clinical trial of the inactivated rotavirus vaccine.
In order to further evaluate the safety and immunogenicity of the inactivated rotavirus vaccine and subsequent vaccine efficacy, it will be more systematically considered and arranged in the design of future clinical trials.

Reviewer 2 Report
Comments and Suggestions for Authors
The authors have systematically revised the manuscript. I have the following remaining comments:
Definition of AE and SEA: thank you for the explanation provided on page 2 of the cover letter; I suggest to make reference to the Good Clinical Practice ICH document and/or add a short explanation in the text.
Definition of events, samples, subjects, occurrences (Tables 1, Tables S1 and S2): occurrence and subject is clear now, is there a difference between events and occurrences (for instance Table 1, High Dose, 1 out of 24 persons had an AE, and this occurred two times in the assessment period, is this correct?). – I am still not sure about the meaning of “correlation” and “irrelevance”; how was it decided whether or not an occurrence was assigned irrelevant or not?
Line 160, low temperature refrigerator: I presume this means a freezer at either -20 or -70 ºC?
Lines 336-338; thank you for your answer; I interpret this in the following way: the fact that a higher AE incidence was detected in the low dose group compared to the three other groups is (only) a statistical effect but technically either not explainable or irrelevant.
Author Response
For research article
|
Response to Reviewer 2 Comments (Round 2)
|
- Summary
Response to the comment:
We thank the Reviewer 2 for the second careful reading and thoughtful comments on previous draft. The comments are all valuable and were very helpful in revising and improving our paper, as well as for providing important guidance regarding our research.
|
2. Questions for General Evaluation |
Reviewer’s Evaluation |
Response and Revisions |
|
Does the introduction provide sufficient background and include all relevant references? |
Yes |
|
|
Is the research design appropriate? |
Yes |
|
|
Are the methods adequately described? |
Yes |
|
|
Are the results clearly presented? |
Can be improved |
|
|
Are the conclusions supported by the results? |
Yes |
|
- Point-by-point response to Comments and Suggestions for Authors
Reviewer comment 1: Definition of AE and SEA: thank you for the explanation provided on page 2 of the cover letter; I suggest to make reference to the Good Clinical Practice ICH document and/or add a short explanation in the text.
Response to the comment 1:
Thank you very much for this suggestion. We have added descriptions of the AE and SAE judgment criteria with reference source descriptions (GCP ICH) and reference in section 2.2. Clinical trial design (line 126-130).
We have added the following description: The International Council for Harmonisation of Technical Requirements for Pharmaceuticals for Human Use (ICH) has published numerous guidelines, including the ICH E6 Good Clinical Practice (GCP) guideline provides standards for reporting of clinical trials and definition of AE and SAEs.
Reviewer comment 2: Definition of events, samples, subjects, occurrences (Tables 1, Tables S1 and S2): occurrence and subject is clear now, is there a difference between events and occurrences (for instance Table 1, High Dose, 1 out of 24 persons had an AE, and this occurred two times in the assessment period, is this correct?). – I am still not sure about the meaning of “correlation” and “irrelevance”; how was it decided whether or not an occurrence was assigned irrelevant or not?
Response to the comment 2:
That is correct, thank you. In high-dose group, 1 out of 24 persons had an AE, and this occurred two times in the assessment period. In high-dose group, N=24, 1 out of 24 (N=1) had AE for 2 times (No. of events=2).
In general reactions, when a subject develops an AE that is consistent with the nature, severity, and frequency of adverse reactions or events mentioned in the current investigator's manual, trial protocol, or product insert, it is reasonable to assume that it is related to the experimental vaccine. Such as fever and localized redness and swelling, which may be accompanied by a combination of symptoms such as general malaise, lethargy, loss of appetite, and malaise.
Among the abnormal reactions, when a subject develops an AE whose nature, severity, and frequency are not mentioned in the current investigator's manual, trial protocol, or product specification, but which causes a certain degree of damage to the subject's body tissues, organs, and functions, it is judged in principle to be a suspected abnormal reaction to preventive vaccination, which may be related to the test vaccine, and which needs to be investigated and judged by the county or municipal expert group on the investigation and diagnosis of abnormal reactions to preventive vaccination.
Adverse event assessment correlation notes have also been appended to the supplementary material (S8. Assessment of adverse events).
Reviewer comment 3: Line 160, low temperature refrigerator: I presume this means a freezer at either -20 or -70 ºC?
Response to the comment 3:
Thank you for this question. Your suggestion is correct. After incubation, plates were placed in -70ºC refrigerator for freeze.
Reviewer comment 4: Lines 336-338; thank you for your answer; I interpret this in the following way: the fact that a higher AE incidence was detected in the low dose group compared to the three other groups is (only) a statistical effect but technically either not explainable or irrelevant.
Response to the comment 4:
Thank you for this professional question. In this regard, we considered that the difference in the number of AE statistics that occurred during the phase I clinical trial was not an objective response to the correlation between vaccine dose and adverse reactions due to the small number of experimental enrollees in each dose group (N=24).
In subsequent clinical trials, we will expand the experimental population, and the safety of the vaccine will continue to be evaluated.
